

# Measuring the developmental function of peer review: a multi-dimensional, cross-disciplinary analysis of peer review reports from 740 academic journals

Daniel Garcia-Costa[1], Flaminio Squazzoni[2], Bahar Mehmani[3] and Francisco Grimaldo[1]

[1] Department of Computer Science, University of Valencia, Valencia, Spain
[2] Department of Social and Political Sciences, University of Milan, Milan, Lombardy, Italy
[3] STM Journals, Elsevier, Amsterdam, The Netherlands

## ABSTRACT

Reviewers do not only help editors to screen manuscripts for publication in academic journals; they also serve to increase the rigor and value of manuscripts by constructive feedback. However, measuring this developmental function of peer review is difficult as it requires fine-grained data on reports and journals without any optimal benchmark. To fill this gap, we adapted a recently proposed quality assessment tool and tested it on a sample of 1.3 million reports submitted to 740 Elsevier journals in 2018–2020. Results showed that the developmental standards of peer review are shared across areas of research, yet with remarkable differences. Reports submitted to social science and economics journals show the highest developmental standards. Reports from junior reviewers, women and reviewers from Western Europe are generally more developmental than those from senior, men and reviewers working in academic institutions outside Western regions. Our findings suggest that increasing the standards of peer review at journals requires effort to assess interventions and measure practices with context-specific and multi-dimensional frameworks.

## INTRODUCTION

Peer review is key for public trust in science (*Bornmann, 2011*). Vetting scientific claims from authors who can often be over-confident and biased towards their own findings before publication is one of the main functions of academic journals. This ensures that only rigorous research reaches public visibility and informs medical treatment, technology innovations and public decisions (*Kharasch et al., 2021*). However, by ensuring high standards of review reports, journals also contribute to improve the value of manuscripts, so enhancing mutual learning between experts (*Rigby, Cox & Julian, 2018*). These two functions of peer review can be called: "quality screening" and "developmental" function (*Lewin, 2014*; *Seeber, 2020*; *Akbaritabar, Stephen & Squazzoni, 2022*).

While the 'publish or perish' academic culture and obsession for rapid dissemination of scientific findings are posing several challenges to peer review (*Edwards & Siddhartha,*

Corresponding author
Francisco Grimaldo,
francisco.grimaldo@uv.es

*2017*), including the recent impact of the fast track publication of COVID-19 pandemic research (*Squazzoni et al., 2021b*; *Horbach, 2021*; *Sullivan et al., 2022*), there is no consensus on how to measure the standards of peer review. While research on attitudes, practices and writing styles of peer reviewers has recently grown thanks to the availability of original data from Publons, various open peer review repositories or single journals (*Casnici et al., 2017*; *Buljan et al., 2020*; *Wolfram, Wang & Abuzahra, 2021*; *Stephen, 2022*; *Rice et al., 2022*; *Thelwall, 2022*), only in a few cases, data include full information on manuscripts and reviewers from different journals and research areas (*Squazzoni et al., 2020*). Furthermore, measuring the quality of peer review is difficult (*Cowley, 2015*). Indeed, efforts have been made to measure the quality of review reports since the end of the 1990s in biomedical research (*Van Rooyen, Black & Godlee, 1999*; *Jefferson, Wager & Davidoff, 2002*; *Van Rooyen, 2001*; *Schroter et al., 2004*; *Schroter et al., 2006*), especially as a means to estimate the efficacy of interventions, *e.g.*, reviewer training. However, there is no systematic measurement of peer review standards that can help assess the state-of-the-art of peer review in various areas of research. This does not permit a rigorous assessment of innovations on peer review at journals, thus undermining the adoption of an evidence-based approach to peer review reforms (*Squazzoni et al., 2020*).

In order to fill this gap, *Superchi et al. (2020)* have recently proposed Arcadia (Assessment of Review reports with a Checklist Available to eDItors and Authors), a tool to assess the quality of peer review reports in biomedical research. By surveying 446 biomedical editors and authors, they identified a checklist for the quality of review reports including five domains as follows: Importance of the study (*i.e.*, the contribution and relevance of the study); robustness of the study methods (*i.e.*, the soundness of the study methods); interpretation and discussion of the study results (*i.e.*, coherence of the study conclusions compared to research questions, external validity and study limitations); reporting and transparency of the manuscript (*i.e.*, data sharing, report guidelines and reproducibility); and characteristics of the peer reviewer's comments (*i.e.*, clarity, objectivity and constructiveness of reviewer comments). They defined quality as: "the extent to which a peer review report helps, first, editors make an informed and unbiased decision about the manuscripts' outcome and, second, authors improve the quality of the submitted manuscript", thus combining both functions of peer review, *i.e.*, quality screening and developmental function.

While this study has improved our understanding of peer review compared to previous research (*Superchi, González & Solá, 2019*), especially in terms of external validation of measurements, their sample of respondents was limited to biomedical experts and the validation test was only subjective, *i.e.*, reflecting opinion of experts rather than current practices (*Pranić et al., 2021*). Given that practices, norms and models of peer review are heterogeneous and field-specific (*Horbach & Halffman, 2018*; *Merriman, 2020*), there is no optimal benchmark to assess current practices and behaviors of peer review across different areas of research. Here, extending measurements to various research areas is key to increase comparability and provide a baseline for future research.

To fill this gap, we have adapted Arcadia and tested it against a rich database of 1.3 million review reports from 740 Elsevier journals. Data from Elsevier journal management

systems were further enriched with data on reviewers and journals from Scopus and other sources. Our aim was to use Arcadia to examine the developmental function of peer review by including multi-dimensional measurements and developing a score that would permit systematic comparisons of peer review reports from different research areas.

We first translated Arcadia into a vocabulary to map the text of peer review reports. This allowed us to provide a multi-dimensional developmental score to compare and assess reports per area of research and reviewer characteristics. We guessed each reviewer's gender, and reconstructed their seniority and geographical/institutional location. We classified journals in quartiles of impact factor using Web of Science. This was to estimate the effects of various factors, either reviewer, field, or journal specific, on report standards. Rather than using humans to rate the quality dimensions of peer review as in previous research (*Superchi et al., 2020*), we used data to measure the current standards and practices of reporting in various journals (*Bianchi, Grimaldo & Squazzoni, 2019*). While the concept of 'quality' is hard to quantify due to its complexity and the co-existence of various goals and stakeholders, measuring standards of reports by means of natural language processing techniques on contents can help us to consider multi-dimensional factors without restricting the observation sample for the sake of human raters (*Ghosal et al., 2022*).

## MATERIALS AND METHODS

### Dataset

Data access required a confidential agreement to be signed on 12th May 2020 between Elsevier and each author of this study. The agreement was inspired by the PEERE protocol for data sharing and included anonymization, privacy, data management and security policies jointly determined by all partners (*Squazzoni, Grimaldo & Marusic, 2017*).

The whole dataset included 1,331,941 reviewer reports from 740 Elsevier journals in all areas of research: Life sciences (hereafter, LS), physical sciences (hereafter, PS), health and medical sciences (hereafter, HMS), and social sciences and economics (hereafter, SSE). Reports referred to the first round of peer review and were related to research manuscripts. Thanks to an ex-post integration and data enrichment from multiple sources, including Elsevier journal data, Scopus for additional information on reviewers and Web of Science for information on journals, we inferred each reviewer's gender, seniority, and country of affiliation. We also had information on the final editorial decision associated with each manuscript, the report time, and the review status. Given the relatively few cases of journals listed among the fourth quartile of impact factor and for the sake of our analysis, we decided to merge Q4 and non-indexed journals in the same category.

Tables 1, 2 and 3 show the number of journals and reports per area of research, journal quartile and reviewers' geographical location. Table 4 shows that women ensured only about 22% of reports, confirming recent findings on the weak involvement of women as reviewers (*Helmer et al., 2017*; *Publons, 2018*; *Stockemer, 2022*).

Each review report was cleaned and standardized by converting to lowercase, removing all non-alphanumerical characters, standardizing breaklines and separator characters and

**Table 1 Number of journals per quartile of impact factor and area of research.**

|  | PS | SSE | HMS | LS | Total |
|---|---|---|---|---|---|
| Journals | 333 | 99 | 174 | 134 | 740 |
| Journals Q1 | 161 | 45 | 40 | 38 | 283 |
| Journals Q2 | 110 | 20 | 40 | 49 | 219 |
| Journals Q3 | 29 | 17 | 32 | 27 | 105 |
| Journals Q4 | 8 | 3 | 7 | 3 | 21 |
| Journals NI | 25 | 14 | 55 | 18 | 112 |

**Table 2 Number of reviews per journal quartile and area of research.**

|  | PS | SSE | HMS | LS | Total |
|---|---|---|---|---|---|
| Reviews | 825.247 | 171.070 | 150.296 | 185.328 | 1.331.941 |
| Reviews Q1 | 602.763 | 146.088 | 51.860 | 88.089 | 888.800 |
| Reviews Q2 | 165.506 | 18.422 | 40.733 | 61.104 | 285.765 |
| Reviews Q3 | 29.743 | 5.147 | 46.596 | 26.375 | 107.861 |
| Reviews Q4 | 2.104 | 468 | 3.236 | 978 | 6.786 |
| Reviews NI | 25.131 | 945 | 7.871 | 8.782 | 42.729 |

**Table 3 Number of reviews per reviewers' geographical location and area of research.** (Note: Countries are classified according to ISO 3166 country codes, while their aggregation complies with the United Nation M49 standard).

|  | PS | SSE | HMS | LS | Total |
|---|---|---|---|---|---|
| Northern America | 120392 | 64254 | 52027 | 52763 | 289436 (21.73%) |
| Western Europe | 64603 | 16539 | 14798 | 17923 | 113863 ( 8.55%) |
| Eastern Asia | 290125 | 32140 | 20583 | 37496 | 380344 (28.56%) |
| Southern Asia | 57994 | 3880 | 6450 | 7124 | 75448 ( 5.66%) |
| Northern Europe | 46505 | 16048 | 13235 | 12387 | 88175 ( 6.62%) |
| Eastern Europe | 37165 | 1722 | 3622 | 5935 | 48444 ( 3.64%) |
| Latin America and the Caribbean | 34886 | 2791 | 6329 | 11713 | 55719 ( 4.18%) |
| Southern Europe | 85495 | 11726 | 15388 | 21733 | 134342 (10.09%) |
| South-East Asia | 19079 | 2158 | 1995 | 3378 | 26610 ( 2.00%) |
| Western Asia (Middle East) | 27071 | 5211 | 5653 | 5121 | 43056 ( 3.23%) |
| Australia and New Zealand | 24925 | 12463 | 5716 | 5756 | 48860 ( 3.67%) |
| Northern Africa | 8006 | 317 | 2300 | 1383 | 12006 ( 0.90%) |
| Central Asia | 306 | 16 | 13 | 35 | 370 ( 0.03%) |
| Sub-Saharan Africa | 5254 | 750 | 955 | 1201 | 8160 ( 0.61%) |
| Micronesia | 134 | 24 | 74 | 52 | 284 ( 0.02%) |
| Melanesia | 70 | 10 | 7 | 17 | 104 ( 0.01%) |
| Polynesia | 23 | 3 | 2 | 5 | 33 ( 0.00%) |
| Missing | 3214 | 1018 | 1149 | 1306 | 6687 ( 0.50%) |

**Table 4   Number of reviews per gender, seniority and area of research.**

|  | PS | SSE | HMS | LS | Total (%) |
|---|---|---|---|---|---|
| Women | 148807 | 44927 | 41754 | 59562 | 295050 (22.15%) |
| Men | 645547 | 120529 | 106096 | 121488 | 993660 (74.60%) |
| Missing gender | 30893 | 5614 | 2446 | 4278 | 43231 (3.25%) |
| <5 years | 21365 | 9463 | 3867 | 4070 | 38765 (2.91%) |
| 5 to 18 years | 435892 | 101008 | 67912 | 85074 | 689886 (51.80%) |
| >18 years | 335270 | 51557 | 69659 | 86483 | 542969 (40.77%) |
| Missing seniority | 32720 | 9042 | 8858 | 9701 | 60321 (4.53%) |

removing repeated white spaces, converting webpage links and reference citations to tokens, removing stop words and words stemming only from the root of each word. Note that after estimating the length of each report, we decided to remove outliers to avoid biasing our analysis. The final dataset included 1,331,941 review reports.

## Standard measurements

In order to estimate peer review standards, we started from Arcadia, a recently released checklist to assess the quality of peer review reports in biomedical research (*Superchi et al., 2020*). Arcadia considers five domains and 14 items, including: Contribution; Relevant literature; Study methods; Statistical methods; Study conclusions; Study limitations; Applicability and generalizability; Study protocol; Reporting; Presentation and organization; Data availability; Clarity; Constructiveness; and Objectivity.

However, considering the specific purposes of Arcadia and its focus restricted to biomedical journals, we added modifications necessary to reflect the characteristics of our dataset, including journals from different areas of research. After translating items into words and running some preliminary test, we decided to merge 'Reporting' with 'Applicability and generalizability' and separate 'Presentation' from 'Organization'. We extracted 'Clarity' by means of readability metrics and decided to disregard 'Constructiveness and objectivity' because these dimensions were hardly quantifiable in our dataset.

This led us to concentrate on the following developmental dimensions:

- **Impact**, *i.e.*, comments from reviewers on the impact of findings or any other manuscript content on society, the economy or whatever external stakeholders, and the study contribution.
- **Relevant literature (literature)**, *i.e.*, comments of reviewers concerning the state-of-the-art and the manuscript references.
- **Study methods (methods)**, *i.e.*, comments from reviewers on materials, methods, and the study design.
- **Statistical methods (statistics)**, *i.e.*, comments from reviewers regarding the statistical content of the study.
- **Study conclusions (conclusions)**, *i.e.*, comments from reviewers on results and conclusions.
- **Limitations**, *i.e.*, comments from reviewers regarding study limitations.

- **Applicability**, comments from reviewers concerning the applicability, generalizability and reproducibility of the study.
- **Presentation**, *i.e.*, comments from reviewers about the presentation of the manuscript, and the quality/readability of tables, figures, and other visualizations.
- **Data availability (data)**, *i.e.*, comments from reviewers regarding data availability.
- **Organization and writing (writing)**, *i.e.*, comments from reviewers about the organization and the linguistic content and style of writing of the manuscript.

## Dictionary building

In order to build a dictionary and also given the characteristics of our dataset, we decided to follow a semi-automatic dictionary building approach, which mostly ensured similar results to manually built dictionaries (*Muresan & Klavans, 2002*; *Godbole et al., 2010*; *Deng et al., 2017*; *Deng et al., 2019*; *Mpouli, Beigbeder & Largeron, 2020*). Given the very large corpus of textual data and the possibility of relying on a predefined list of developmental dimensions extracted from Arcadia, we used manual checks on the output of each iteration to verify the process and minimize possible mistakes.

We followed five steps: (1) corpus creation, (2) pre-processing and cleaning, (3) vector representation of the corpus, (4) term extraction and (5) validation, which included steps 4 and 5 to be repeated several times (see the full process in Fig. 1). In step 2, we converted the text into lower case, removed non-alphanumeric characters, trimmed white spaces and line breaks, tokenised web links and citations, removed stop words and finally applied stemming to standardize words. In step 3, we built an unsupervised Word2Vec model using the H2O API (https://www.h2o.ai/) in R (https://www.r-project.org/) to create a vector representation of our corpus. We departed from an initial list of manually defined terms by revising a sample of review reports and selecting ten terms for each dimension (see Table 5). By using bigrams, we minimised context-specific ambiguities while categorizing individual words.

In step 4, we used the Word2Vec model to search for near terms in all review texts. We extracted new terms by running the method 'findSynonyms' from the H2O API and selected the most frequent similar terms (*i.e.*, those with a normalized score, returned by this method, higher than 0.75) and listed among list candidates. The identification of non-existing unigram and bigram terms required different procedures: whenever a new bigram term was selected, we checked if any of its words already existed as unigram terms and, if so, the term was dropped out.

In step 5, we validated the list of new terms. We used a KWIC method to validate each new term, by checking the context in which the term was used throughout the corpus, obtaining some examples and assessing whether the term was appropriate or not. Given that this was context-dependent, we opted for a manual validation performed by a male PhD student (val1) and a female Master degree student (val2), with a male senior researcher (val3) decisive in case of any conflicting assessments. Note that these were all domain experts. During such a validation step, these experts were allowed to manually check when an unigram was dropped out due to its ambiguity by reconstructing the context and eventually converting the unigram to the correct bigram.
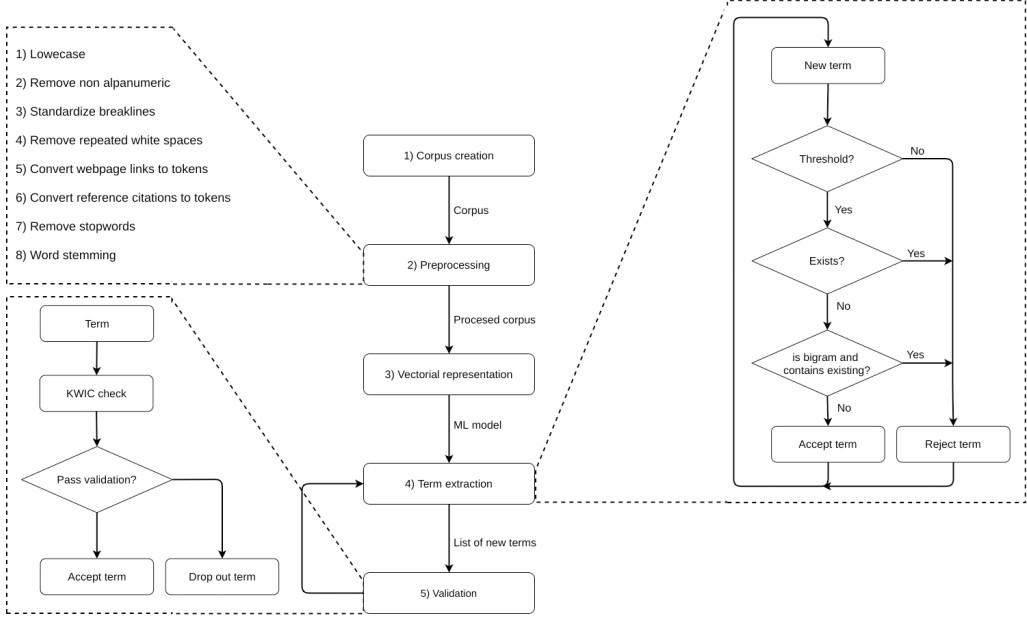

**Figure 1** **Steps of the dictionary building process.**

This allowed us to use the output list of terms from the previous iteration as input for each new iteration. Any extraction and validation step was repeated until all new terms had low frequency values. This allowed us to obtain a total of 1,565 terms (see Table 6 for the distribution of terms of each dimension of the developmental score).

These final list of terms was then used to build a LIWC (Linguistic Inquiry and Word Count) (https://liwc.wpengine.com/) style dictionary. While our dictionary could be used in LIWC or any other program or library which accepts LIWC style dictionaries, here we used the package "quanteda.dictionaries" (https://github.com/kbenoit/quanteda.dictionaries) to estimate the developmental values for each review report in our dataset. These values reflected the number of words found from each category in the text reports.

## Developmental score

Given that the distribution of developmental terms followed a Zipfian distribution with discrete different scales, aggregating all dimensions into a single score required to avoid that any specific dimension would predominate over others. To avoid this, we normalized each dimension by using the empirical cumulative distribution function (ECDF), thus transforming the discrete word count values into a real scale between 0 and 1. We used the arithmetic mean of these standardized values to aggregate them and generated a unique score for each report. Whenever a report did not contain any word from a certain dimension, its assigned value was 0.

The calculus of the score followed this formula:

$$Score = \frac{1}{n} * \sum_{i=1}^{n} F_{D_i}(v_i)$$

**Table 5  Initial seed terms for each developmental dimension.**

| Developmental dimensions | Initial seed terms |
|---|---|
| Impact | relevant, impact, novel, original, innovator paper, interest paper, disappointing paper, important topic, relevant paper, research community |
| Relevant Literature | cite, consider reference, require reference, reference paper, related work, literature, bibliography, similar work, previous work, existing work |
| Study Methods | methodology, approach, experiment, techniques, analysis, procedures, provide justification, provide comparison, exploratory, meticulous |
| Statistical Methods | statistics, null hypothesis, regression, coefficient, significance, correlation, deviation, Bayesian, response variable, effect size |
| Study Conclusions | result, discussion, conclusion, findings, research question, unjustified, evidence, inconsistency, unsolved problem, explanation |
| Limitations | limitations, weakness, robustness, future work, lack acknowledg, acknowledg limit, expertise, under-investigated, flaws, bottleneck |
| Applicability | work applicability, application domain, reproducible, generalizable results, generalizable study, scalable, transferable, irreproducible, reusable, universal method |
| Presentation | table, figure, row, column, image, axis, caption, legend, graph, footer |
| Data Availability | database, data available, accessible data, experiment data, publish data, repository, source code, opaque, secrecy, available resources |
| Organization and Writing | rewrite, well written, poor written, reorganize, move, spelling, page, line, sentence, paragraph |

**Table 6  Number of terms for each dimension of the developmental score.**

| Item | Num of terms | Item | Num of terms |
|---|---|---|---|
| Impact | 175 | Literature | 235 |
| Methods | 240 | Statistics | 122 |
| Conclusions | 283 | Limitations | 71 |
| Applicability | 139 | Presentation | 72 |
| Data | 128 | Writing | 168 |

where $F_{D_i}$ was the cumulative distribution function of the discrete variable, while $D_i$ was the Zipfian variable starting in 1. Note that in case $F_{D_i}(v_i) = 0$, no term was found in the report regarding a given dimension. Figure 2 shows the distribution of our developmental score.
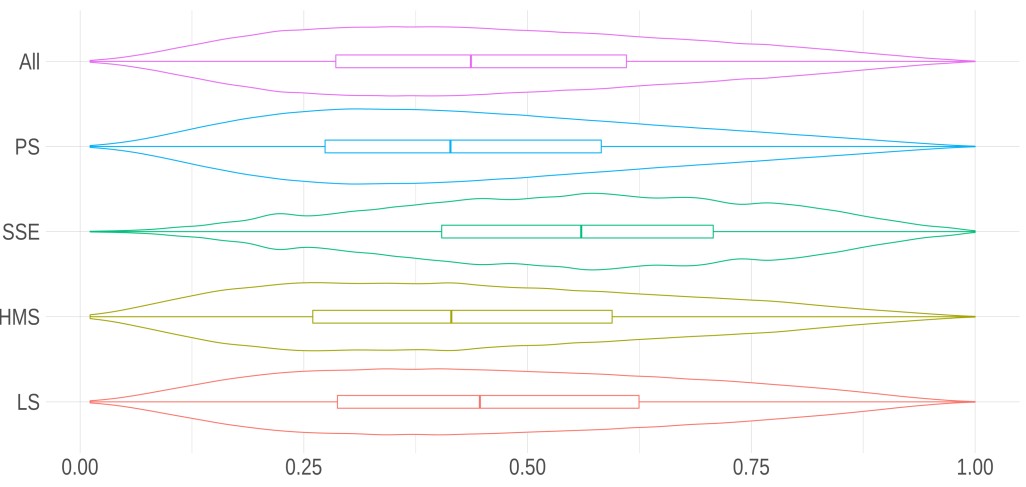

**Figure 2** **Distribution of the developmental score per research area.** The density curves in this violin plot show the distribution of the score for all research areas and, separately, for PS, SEE, HMS or LS.

**Table 7** **Explained variance by each principal component.**

|      | % of variance | Cumulative % of variance |
| ---- | ------------- | ------------------------ |
| PC1  | 39.38         | 39.38                    |
| PC2  | 9.88          | 49.26                    |
| PC3  | 8.76          | 58.02                    |
| PC4  | 7.18          | 65.20                    |
| PC5  | 6.99          | 72.19                    |
| PC6  | 6.63          | 78.82                    |
| PC7  | 5.99          | 84.81                    |
| PC8  | 5.54          | 90.36                    |
| PC9  | 5.24          | 95.59                    |
| PC10 | 4.41          | 100.00                   |

## Score internal validity

By using the R package FactoMineR, we performed a principal component analysis to check the amount of variance for each dimension. Table 7 shows that up to nine principal components were needed to explain at least 95% of our variance. This indicates that there was no correlation between our dimensions, so confirming the main finding from Arcadia (*Superchi et al., 2020*).

## Internal consistency

In order to assess the internal consistency of our developmental score, we estimated Cronbach's alphas and the total-item correlation by using the R package Psych. Note that that threshold of acceptance should be greater than 0.70 for alpha, while item-total correlation should be greater than 0.30. Table 8 shows that we achieved a global Cronbach alpha of 0.82. This indicates that there is no developmental dimension that could have

**Table 8  Cronbach alphas and Item-total correlations.**

| Dimension | α if item was dropped | Item-total correlation |
|---|---|---|
| Impact | 0.81 | 0.53 |
| Relevant literature | 0.82 | 0.47 |
| Study Methods | 0.81 | 0.60 |
| Statistical Methods | 0.80 | 0.63 |
| Study Conclusions | 0.79 | 0.72 |
| Limitations | 0.81 | 0.55 |
| Applicability | 0.81 | 0.58 |
| Presentation | 0.82 | 0.43 |
| Data availability | 0.82 | 0.49 |
| Organization and writing | 0.80 | 0.65 |

**Table 9  CFA factor loadings for each developmental item.**

| Indicator | Estimate | Std.Err | P( > |z|) |
|---|---|---|---|
| Impact | 1.00 | 0.00 | 0.00 |
| Relevant literature | 1.01 | 0.00 | 0.00 |
| Study methods | 1.10 | 0.00 | 0.00 |
| Statistical methods | 1.19 | 0.00 | 0.00 |
| Study conclusions | 1.28 | 0.00 | 0.00 |
| Limitations | 1.19 | 0.00 | 0.00 |
| Applicability | 1.20 | 0.00 | 0.00 |
| Presentation | 0.88 | 0.00 | 0.00 |
| Data availability | 1.06 | 0.00 | 0.00 |
| Organization and writing | 1.13 | 0.00 | 0.00 |

been dropped that would have increased the value of alpha. Note also that the item-total correlation for each dimension was greater than the recommended minimum value of 0.30. This test demonstrates that our developmental dimensions were consistent throughout the whole sample, without any dimension biasing our measurements.

We also applied an additional method to evaluate consistency, *i.e.*, the Confirmatory Factor Analysis (CFA). Our CFA showed a good fit between model and data, with a CFI value of 0.93, which was greater than the recommended minimum of 0.90, and a RMSEA of 0.07, which was smaller than the recommended maximum of 0.08. As regards coefficients, note that all developmental items had significant *p*-values (see Table 9).

## Gender guessing

Gender was guessed as previously described in *Squazzoni et al. (2021b)*. Specifically, we queried the Python package gender-guesser about the first names and countries of origin, if any. Gender-guesser allowed us to minimize gender bias and achieve the lowest misclassification rate (less than 3% for Benchmark 1 in *Santamaría & Mihaljević (2018)*). For names classified by gender-guesser as 'mostly_male', 'mostly_female', 'andy' (androgynous) or 'unknown' (name not found), we used GenderAPI (https://gender-api.com/), which

ensures that the level of mis-classification is around 5% (see Table 4 in *Santamaría & Mihaljević (2018)*) and has the highest coverage on multiple name origins (see Table 5 in *Santamaría & Mihaljević, 2018*). This procedure allowed us to guess the gender of 94.5% of academics in our sample, 45.1% coming from gender-guesser and 49.2% from GenderAPI. The remaining 5.5% of academics were assigned an unknown gender. Note that this level of gender guessing is consistent with the non-classification rate for names of academics in previous research (*Santamaría & Mihaljević, 2018*). Note also that while we were aware that any gender binary definition did not adequately represent non-binary identities, to the best of our knowledge, there was no better instrument to guess gender for such a large pool of individuals.

### Seniority

Reviewer seniority was estimated by using the number of years since their first publication record in the Scopus database. This information was retrieved through the Elsevier International Center for the Study of Research (ICSR Lab) computational platform. We used either the Scopus ID, the e-mail address or the full name plus country (in this order of preference) to find a unique matching profile in the Scopus database. We followed a conservative rule and reviewers without a profile in Scopus or with more than a single matching profile (*i.e.*, not being uniquely identifiable) were excluded from the analysis, whenever using seniority as a variable. By following *Squazzoni et al. (2021a)*, we assumed that first publications would correspond to the period in which reviewers were completing their MD or PhD. We then considered a cut-off of 18 years to identify junior vs. senior reviewers, *i.e.*, full professors.

## RESULTS

### Developmental score

Figure 2 shows that peer review reports submitted to social sciences and economics (SEE) journals showed the highest developmental standards compared to all areas of research. Table 10 shows that SSE reports had the highest scores in all developmental dimensions except for *Presentation*, for which they scored lower than reports from any other area of research. We used a Gamma Generalized Linear Model to analyze the relation with relevant covariates since the developmental score fits this family of distributions (as reported by Generalized Additive Models for Location, Scale and Shape (https://www.gamlss.com/)). Table 11 indicates that the differences in the developmental standards of peer review between areas of research were on average around 10%, with remarkable heterogeneity.

Except for SSE, journals with highest impact factors generally showed higher developmental standards of reports (see Fig. 3). It is interesting to note that the standards of reports in PS and LS did not seem to reflect impact factor hierarchies, as developmental scores were more stable across the first three quartiles than in any other research areas. Interestingly, in SSE journals with a higher impact factor did not show the highest report standards: journals listed among the second and third quartiles in the ranking of impact factor of economics and social science journals had relatively higher standards compared to high-ranked journals (see Fig. 3).

**Table 10   Mean and standard deviation (in brackets) for each developmental score dimension per research area.**

|  | PS | SSE | HMS | LS |
|---|---|---|---|---|
| Impact | 0.473 (0.318) | 0.662 (0.317) | 0.516 (0.329) | 0.521 (0.325) |
| Literature | 0.377 (0.371) | 0.509 (0.384) | 0.328 (0.362) | 0.358 (0.371) |
| Methods | 0.527 (0.316) | 0.585 (0.31) | 0.43 (0.314) | 0.479 (0.313) |
| Statistics | 0.442 (0.329) | 0.645 (0.329) | 0.499 (0.338) | 0.484 (0.335) |
| Conclusions | 0.487 (0.302) | 0.608 (0.303) | 0.521 (0.306) | 0.559 (0.307) |
| Limitations | 0.369 (0.361) | 0.608 (0.364) | 0.437 (0.372) | 0.405 (0.375) |
| Applicability | 0.441 (0.361) | 0.532 (0.359) | 0.423 (0.366) | 0.447 (0.369) |
| Presentation | 0.42 (0.36) | 0.314 (0.331) | 0.322 (0.342) | 0.407 (0.365) |
| Data | 0.315 (0.364) | 0.512 (0.39) | 0.38 (0.377) | 0.388 (0.376) |
| Writing | 0.507 (0.31) | 0.543 (0.305) | 0.492 (0.314) | 0.556 (0.313) |

**Table 11   Effect of research area and journal impact factor on the developmental score using a Gamma Generalized Linear Model with developmental score as response variable.**

|  | Dependent variable: |
|---|---|
|  | Developmental score |
| AreaHMS | $-0.071^{***}$ (0.001) |
| AreaPS | $-0.105^{***}$ (0.001) |
| AreaLS | $-0.084^{***}$ (0.001) |
| IFQuartileQ2 | $0.033^{***}$ (0.002) |
| IFQuartileQ3 | $0.058^{***}$ (0.004) |
| IFQuartileQ4+NI | $-0.025^{***}$ (0.006) |
| AreaHMS:IFQuartileQ2 | $-0.047^{***}$ (0.003) |
| AreaPS:IFQuartileQ2 | $-0.052^{***}$ (0.002) |
| AreaLS:IFQuartileQ2 | $-0.035^{***}$ (0.002) |
| AreaHMS:IFQuartileQ3 | $-0.158^{***}$ (0.004) |
| AreaPS:IFQuartileQ3 | $-0.069^{***}$ (0.004) |
| AreaLS:IFQuartileQ3 | $-0.052^{***}$ (0.004) |
| AreaHMS:IFQuartileQ4+NI | $-0.060^{***}$ (0.007) |
| AreaPS:IFQuartileQ4+NI | $-0.037^{***}$ (0.007) |
| AreaLS:IFQuartileQ4+NI | $-0.027^{***}$ (0.007) |
| Constant | $0.547^{***}$ (0.001) |
| Observations | 1,331,247 |
| Log Likelihood | 196,834.500 |
| Akaike Inf. Crit. | $-393,636.900$ |

Notes.
$^{*}p < 0.1.$
$^{**}p < 0.05.$
$^{***}p < 0.01.$
Reference categories were: reports submitted to SSE journals listed in the first quartile of impact factor.
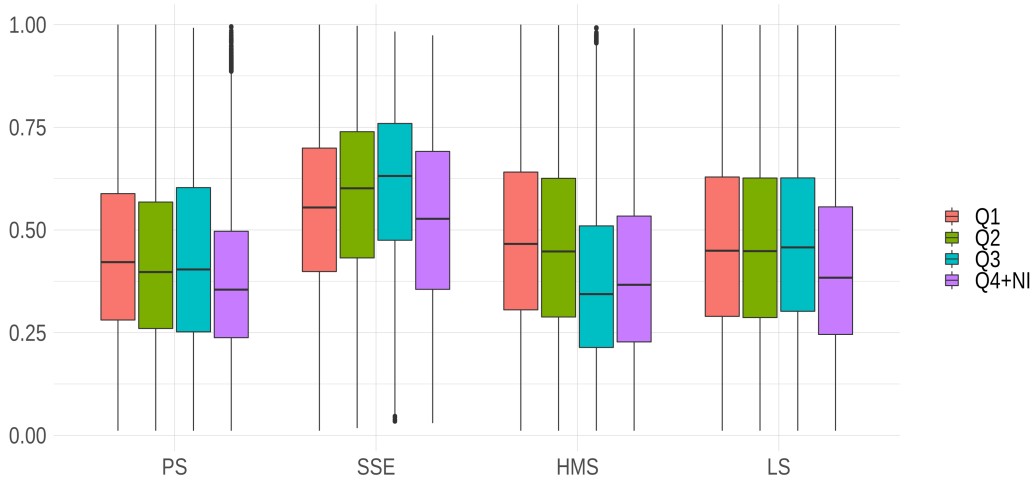

**Figure 3** **Interaction between journal prestige and research area.** Note that due to the restricted number of cases in the sample and for the sake of readability, we included fourth quartile and not-indexed journals in the same category.

**Table 12** **Effect of report delivery time on the developmental score per research area using a Gamma Generalized Linear Model with developmental score as response variable.**

|  | Dependent variable: | | | |
|---|---|---|---|---|
|  | Developmental score | | | |
|  | **PS** | **SSE** | **HMS** | **LS** |
| Report delivery time | 0.002[***] | 0.001[***] | 0.002[***] | 0.003[***] |
|  | (0.00002) | (0.00003) | (0.00005) | (0.00004) |
| Constant | 0.401[***] | 0.525[***] | 0.400[***] | 0.414[***] |
|  | (0.0004) | (0.001) | (0.001) | (0.001) |
| Observations | 824,954 | 171,055 | 149,978 | 185,256 |
| Log Likelihood | 145,774.700 | 21,203.550 | 18,616.660 | 21,472.280 |
| Akaike Inf. Crit. | −291,545.500 | −42,403.110 | −37,229.320 | −42,940.560 |

**Notes.**
[*]$p < 0.1$.
[**]$p < 0.05$.
[***]$p < 0.01$.

However, the higher developmental score of reports seems to come at a price: in SSE journals, the median delivery time of reviewers is 24 days against 15 days for reviewers from HMS, 17 days in LS, and 19 days in PS journals. Table 12 shows a positive correlation between delivery time and developmental score of reports. Although various factors could influence the turn-round time of reports, including editorial standards of reminders, this would suggest a potential trade-off between the developmental content of reports and quick editorial decisions (*Sullivan et al., 2022*).

## Reviewer characteristics

Here, we aimed to examine whether the developmental score of reports could reflect certain reviewer characteristics, such as gender and seniority. When considering reviewer gender,

**Table 13  Effect of gender and seniority on the developmental score per area of research using a Gamma Generalized Linear Model with developmental score as response variable.**

| | Dependent variable: | | | |
|---|---|---|---|---|
| | Developmental score | | | |
| | **PS** | **SSE** | **HMS** | **LS** |
| Seniority 5 to 18 years | −0.050*** | −0.064*** | −0.039*** | −0.033*** |
| | (0.004) | (0.005) | (0.007) | (0.006) |
| Seniority > 18 years | −0.065*** | −0.090*** | −0.058*** | −0.058*** |
| | (0.004) | (0.005) | (0.007) | (0.006) |
| Gender Man | −0.018*** | −0.087*** | −0.089*** | −0.054*** |
| | (0.004) | (0.005) | (0.008) | (0.008) |
| Seniority 5 to 18 years: Gender Man | 0.003 | 0.074*** | 0.026*** | 0.0001 |
| | (0.004) | (0.005) | (0.009) | (0.008) |
| Seniority > 18 years: Gender Man | 0.005 | 0.099*** | 0.020*** | 0.015*** |
| | (0.004) | (0.006) | (0.009) | (0.008) |
| Constant | 0.504*** | 0.630*** | 0.532*** | 0.538*** |
| | (0.003) | (0.005) | (0.007) | (0.006) |
| Observations | 762,864 | 156,575 | 138,933 | 171,641 |
| Log Likelihood | 129,470.100 | 19,139.370 | 17,814.220 | 19,073.790 |
| Akaike Inf. Crit. | −258,928.200 | −38,266.750 | −35,616.450 | −38,135.570 |

Notes.
*$p < 0.1$.
**$p < 0.05$.
***$p < 0.01$.

Reference categories were: women reviewers with < 5 years of seniority. Note that seniority was estimated by looking at the first publication of each reviewer indexed in Scopus.

we did not find any considerable effects on report standards. The only exception were reports submitted to SSE and HMS journals, where reports from women obtained scores approximately 8% higher than those submitted by men (see Table 13). This would confirm recent research reporting weak gender effects on reviewer attitudes, recommendations and writing styles in various research areas and journal contexts (*Bravo et al., 2019*; *Buljan et al., 2020*; *Bolek et al., 2022*).

When considering reviewer seniority (for detail on the measurement of seniority, see the Method Section), we found a difference of 8% between junior and senior reviewers in all research areas. Junior reviewers generally ensure comparatively highest developmental standards of reports (see Table 13). For instance, in SSE journals, reports from juniors scored around 10% higher than those submitted by seniors. While this could simply reflect the fact that seniors would be more concise in their reports or have less time for reviews (*Hochberg, 2010*; *Merrill, 2014*; *Bianchi et al., 2018*), the higher developmental scores of reports from junior scholars could also reflect reputation building strategies, *e.g.*, showing their diligence and reliability to journal editors in view of potential future submissions (*Mahmić-Kaknjo, Utrobičić & Maručić, 2021*).

## Institutional and geographical factors

Here, we aimed to examine whether institutional or geographical factors could influence the developmental score of reports. This was to consider potential heterogeneity in practices and style of reviewing (*Publons, 2018*). Indeed, our results showed considerable variations of the developmental score when controlling for the institutional and geographical embeddedness of reviewers. Although with certain specificities due to research areas, results indicate that reviewers from Western Europe would have higher developmental standards compared to reviewers from other regions, except for reports submitted to HSM and LS journals, though with a very weak statistical difference (about 1%). Table 14 shows that reports submitted by reviewers from Asian regions would be less developmental (10–15% lower than reviewers from Europe).

Figure 4 shows the distribution of the developmental score per dimension and institutional and geographical origins of reviewers. The distribution suggests that report scores were generally higher for writing, conclusions, methods and impact, thus confirming research showing that reviewers would tend to concentrate more preferably on certain aspects of manuscripts (*Siler, Lee & Bero, 2015*; *Herber et al., 2020*; *Teplitskiy et al., 2018*; *Stephen, 2022*). Data and limitations showed lower scores, the latter also showing the greatest variation in the score distribution per region. More importantly, our results showed that reports from reviewers from Northern America scored higher on data and statistics than reports from reviewers from Western Europe and any other region. Note also that reports from reviewers from various regions greatly varied as to how they focused on the way the text of manuscripts was written and organized.

## DISCUSSION AND CONCLUSIONS

Although academic journals have been recently threatened by the need for rapid dissemination of scientific information, their real hallmark is their capacity to maintain rigorous standards of peer review. This is key to ensure that scientific claims can be trusted by the public (*Kharasch et al., 2021*). This has been especially important during the recent pandemic and will also be so in the post-pandemic science (*Bauchner, Fontanarosa & Golub, 2020*; *Palayew et al., 2020*). However, this requires that each report submitted by reviewers meets the highest professional standards, which is also instrumental in maintaining the credibility and legitimacy of journals for authors who submit their manuscripts (*Pranić et al., 2021*).

Our research shows that standards of peer review are robust though with certain field-specific characteristics. The fact that developmental standards of peer review are higher in SSE journals would confirm the specificity of the historical institutional trajectory of peer review in these fields. As suggested by previous research (*Huisman & Smits, 2017*; *Merriman, 2020*), editorial standards of journals in these fields typically include double anonymized peer review and a tendency towards more constructive and elaborated reports. Furthermore, while the debate is open on the editorial standards of top journals in this area of research and their excessive prominence and concentration (*Card & DellaVigna, 2013*; *Teele & Thelen, 2017*; *Akbaritabar & Squazzoni, in press*), our findings would reveal that more specialized

**Table 14** The effect of the geographical location of reviewers on the developmental score per area of research.

| | Dependent variable: | | | |
|---|---|---|---|---|
| | Developmental score | | | |
| | PS | SSE | HMS | LS |
| Southern Asia | −0.133*** | −0.081*** | −0.105*** | −0.170*** |
| | (0.001) | (0.004) | (0.003) | (0.003) |
| Northern Europe | −0.018*** | −0.026*** | 0.014*** | 0.005* |
| | (0.001) | (0.002) | (0.003) | (0.003) |
| Southern Europe | −0.036*** | −0.032*** | −0.036*** | −0.071*** |
| | (0.001) | (0.003) | (0.003) | (0.002) |
| Northern Africa | −0.131*** | −0.158*** | −0.113*** | −0.148*** |
| | (0.002) | (0.009) | (0.004) | (0.005) |
| Sub-Saharan Africa | −0.052*** | −0.167*** | −0.021*** | −0.055*** |
| | (0.003) | (0.006) | (0.007) | (0.006) |
| Latin America and the Caribbean | −0.057*** | −0.083*** | −0.038*** | −0.080*** |
| | (0.001) | (0.004) | (0.003) | (0.003) |
| Western Asia (Middle East) | −0.115*** | −0.059*** | −0.118*** | −0.135*** |
| | (0.001) | (0.003) | (0.003) | (0.003) |
| Australia and New Zealand | −0.041*** | −0.028*** | 0.038*** | 0.011*** |
| | (0.002) | (0.003) | (0.004) | (0.004) |
| Eastern Europe | −0.082*** | −0.083*** | −0.064*** | −0.084*** |
| | (0.001) | (0.005) | (0.004) | (0.003) |
| Northern America | −0.031*** | −0.069*** | 0.001 | −0.013*** |
| | (0.001) | (0.002) | (0.002) | (0.002) |
| South-East Asia | −0.095*** | −0.072*** | −0.055*** | −0.103*** |
| | (0.002) | (0.005) | (0.005) | (0.004) |
| East Asia | −0.145*** | −0.131*** | −0.125*** | −0.169*** |
| | (0.001) | (0.002) | (0.002) | (0.002) |
| Constant | 0.521*** | 0.617*** | 0.467*** | 0.527*** |
| | (0.001) | (0.002) | (0.002) | (0.002) |
| Observations | 821,213 | 169,984 | 148,745 | 183,842 |
| Log Likelihood | 167,148.900 | 23,405.270 | 21,644.070 | 28,294.330 |
| Akaike Inf. Crit. | −334,271.800 | −46,784.550 | −43,262.150 | −56,562.670 |

**Notes.**

Countries are classified according to ISO 3166 country codes, while their aggregation complies with the United Nation M49 standard. In case of Sub-Saharan Africa, more than the 50% of our observations included reviewers located in South Africa). We used a Gamma Generalized Linear Model with developmental score as response variable.

*$p < 0.1$.
**$p < 0.05$.
***$p < 0.01$.

The reference category were Western European reviewers.

or relatively newly established journals are more keen to adopt developmental peer review, with reviewers probably more encouraged to provide constructive and elaborate reports (*Merriman, 2020*). Furthermore, the fact that standards were more homogeneous across PS and LS journals, at least those listed among the first three quartile of impact factor, would suggest that in these fields, there are more consistent standards of evaluation.

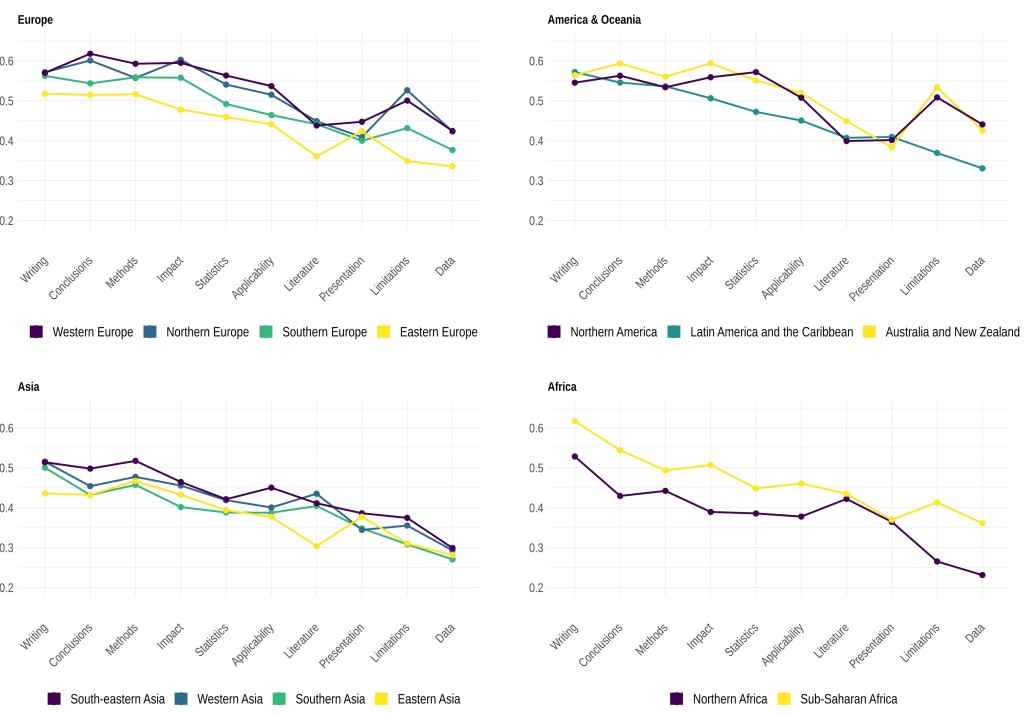

**Figure 4** Median values of each dimension of the developmental score (*i.e.*, cumulative distribution functions $F_{D_i}$ in Materials and Methods) per geographical region.

However, the price to be paid for developmental peer review seems to be a substantial delay in the process, which has always been subject to debate (*Bjrk & Solomon, 2013*). Our results suggest a clear trade-off between developmental peer review and delivery time. This means that, in principle, if reviewers would take more time to deliver their reports generally, this would result in a higher developmental content of reports. However, adding further time to reports in SSE journals would increase the developmental score of reports less than in other areas of research. For instance, we estimated that if reviewers in PS, HMS or LS journals would take ten days more than their median value for report delivery, their expected developmental score would on average increase 3%, thus not reaching the actual median developmental score of SSE reports. Given the recently established fast track options to speed up peer review during the pandemic, it would be interesting to study whether these time pressures have compromised the developmental standards reported here and in which research area (*Horbach, 2021*; *Squazzoni et al., 2021a*).

Our findings indicate that junior scholars are more developmental than more senior reviewers, as are women reviewers in certain fields, such as SSE and HMS, where women reviewers obtained scores slightly higher than men. This would confirm recent findings on relatively weak gender specificities in peer review in various contexts and research areas, including results from linguistic analysis of reports (*Bravo et al., 2019*; *Buljan et al., 2020*; *Squazzoni et al., 2021a*).

We also found evidence that standards reflect geographical and institutional conditions. The report standards are heterogeneous across world regions, while there is an increasing involvement of reviewers from Asian regions compared to less recent data from Publons' state of the art report (*Publons, 2018*). While standards could reflect certain language and cultural specificities and peer review has profound Western historical roots (*Lamont, 2009*), our findings suggest that efforts put forward by publishers and associations regarding higher involvement and inclusion of non-Western academics in journal peer review seem to paid off. However, we must improve on training initiatives and diversity policies to reinforce standards and establish widely shared practices of peer review.

Our findings call for reconsideration of various initiatives on peer review. First, it is important that whenever trying to assess the efficacy of intervention on peer review standards, we use multi-dimensional, context-specific measurements that expand our analysis beyond a few dimensions as in current research, *e.g.*, length of reports (*Publons, 2018*; *Bianchi, Grimaldo & Squazzoni, 2019*). For instance, previous research found that any intervention to improve peer review was relatively unsuccessful in improving the quality of reports (*Jefferson, Wager & Davidoff, 2002*; *Schroter et al., 2004*; *Schroter et al., 2006*; *Bruce, Chauvin & Trinquart, 2016*). Our findings suggest that these results could have been biased by not sufficiently rich, large-scale or systematic measurements of intervention outcomes, or in any case they could have been penalized by lack of appropriate, context-specific benchmarks. While experimental trials are key to assess interventions, measuring peer review only off-line, during specific designed interventions is costly and sometimes limited. Our study suggests that measuring peer review reports more regularly via natural language processing and other machine learning and data science techniques could be a viable alternative to assess internal editorial practices. However, this requires collaboration between publishers, journals and scholars in data sharing initiatives, which are unfortunately only rare (*Squazzoni et al., 2020*).

This given, our study also has certain limitations. First, we used gender guessing techniques, which did not adequately represent non-binary identities, and estimated the seniority of individuals by looking at the number of years since their first record in the Scopus database. However, to the best of our knowledge, there were no better instruments to guess gender and seniority for such a large pool of individuals. Second, our dataset includes only a restricted sample of reports from Elsevier journals in a short time-frame. Although Elsevier does have one of the largest journal portfolios of all publishers, expanding this analysis by including reports from journals from other publishers would be an important step forward. While creating a common database from different publishers is at the moment impossible, due to lack of a data sharing infrastructure solving legal and technical obstacles and creating opportunities for cooperation, a possible extension of our work would be to test our developmental score on available online repositories of peer review reports. Here, considering a longer time-frame could provide a dynamic picture of these standards and not only a cross-sectional comparison.

Furthermore, measuring peer review report standards by looking only at the text of reports separately from the context could provide a rather narrow view of peer review. For instance, each report is linked to others mutually associated with the same manuscript in

that the quality of the process has a complex dimension. In this regard, unfortunately we could control neither for the possible effect of peer review guidelines at the journal level nor for the specific effect of varying peer review models adopted by journals. Although recent research suggests that the peer review model does not dramatically change the way reviewers write their reports (*Bravo et al., 2019*; *Buljan et al., 2020*), the fact that journals can vary greatly on the guidelines to their reviewers (*Seeber, 2020*) could be an interesting subject of investigation. Assessing the effect of these internal policies on the developmental content of reports systematically and comparatively would be indeed a major achievement.

Another point is the role of the context. Peer review is performed in a complex, hyper-competitive and hierarchical academic environment, with great variations in terms of areas of research and institutional contexts where competitive pressures and standards of cooperation greatly differ. In our study, we could not control for these confounding factors, including any author-editor-reviewer competitive/cooperative relationships, which could have important implications on the standards of reports (*Bravo et al., 2018*; *Teplitskiy et al., 2018*; *Dondio et al., 2019*).

Furthermore, while developmental peer review is deeply rooted in the institutional tradition of social sciences (*Lamont, 2009*; *Merriman, 2020*), in other areas of research and for specific type of journals, fast editorial decisions and rapid quality screening of manuscripts could be more relevant, regardless of the impact of exogenous factors such as the COVID pandemic. However, even editorial practices and journal guidelines could influence indirectly the development of manuscripts as authors could adapt their manuscript to potential requests and evaluation standards before submitting them to journals. This implies that drawing a straight line between quality screening and developmental function of peer review can be sometimes difficult. As correctly suggested by *Horbach & Halffman (2018)*, peer review is more than review reports and estimating its dimensions and properties calls for a complex set of factors and processes.

With all these caveats, we believe that concentrating on reports, making dimensions and measurements more transparent, identifying context-specific standards is also instrumental to enhance reviewer training initiatives. Given the higher involvement of non-Western regions and their importance in the changing demography of the scientific community, we must expand the traditional target and audiences of training initiatives, increase their diversity and inclusion, and ensure permanent initiatives rather than on-off programs (*Schroter et al., 2004*). Specifying the various functions of peer review and the required multi-dimensional competence, and establishing more informative and standardized journal guideline would also help to reduce the mismatch of expectations and practices (*Köhler et al., 2020*; *Seeber, 2020*).

## ACKNOWLEDGEMENTS

We gratefully acknowledge the support on data extraction from the IT staff of Elsevier, specifically Ramsundhar Baskaravelu and his team. This work uses Scopus data provided by Elsevier through ICSR Lab (Elsevier International Center for Study of Research). We also thank Dave Santucci from Elsevier Scopus API team and Kristy James from ICSR for

their support on data enrichment about authors and reviewers. We thank Maite Gandia for her preliminary work on analyzing the text of review reports and her help in the first steps of this piece of research. We thank Joan Marsh and Mario Maliki for interesting comments on a preliminary version of the manuscript. Usual caveats apply.

### Funding

Daniel Garcia-Costa and Francisco Grimaldo are supported by the Spanish Ministry of Science, Innovation and Universities (MCIU), the Spanish State Research Agency (AEI) and the European Regional Development Fund (ERDF) under project RTI2018-095820-B-I00. Flaminio Squazzoni is supported by a "Department of Excellence" grant from the Italian Ministry of Education, University and Research to the Department of Social and Political Sciences of the University of Milan, a grant from PRIN-MIUR (Progetti di Rilevante Interesse Nazionale –Italian Ministry of University and Research) (Grant Number: 20178TRM3F001 "14All") and a grant from the University of Milan (Grant Number: PSR2015-17 Transition Grant). The funders had no role in study design, data collection and analysis, decision to publish, or preparation of the manuscript.

### Grant Disclosures

The following grant information was disclosed by the authors:
Spanish Ministry of Science, Innovation and Universities (MCIU).
The Spanish State Research Agency (AEI).
The European Regional Development Fund (ERDF): RTI2018-095820-B-I00.
Italian Ministry of Education, University and Research to the Department of Social and Political Sciences of the University of Milan.
Progetti di Rilevante Interesse Nazionale–Italian Ministry of University and Research: 20178TRM3F001 "14All".
University of Milan: PSR2015-17.

### Competing Interests

Bahar Mehmani is an Elsevier employee. Elsevier provided data for this study through ICSR Lab (Elsevier International Center for Study of Research).

### Author Contributions

- Daniel Garcia-Costa conceived and designed the experiments, performed the experiments, analyzed the data, prepared figures and/or tables, authored or reviewed drafts of the article, and approved the final draft.
- Flaminio Squazzoni conceived and designed the experiments, analyzed the data, authored or reviewed drafts of the article, and approved the final draft.
- Bahar Mehmani conceived and designed the experiments, authored or reviewed drafts of the article, and approved the final draft.

- Francisco Grimaldo conceived and designed the experiments, performed the experiments, analyzed the data, authored or reviewed drafts of the article, and approved the final draft.

**Data Availability**

The data is available at Harvard Dataverse: Daniel Garcia-Costa,; Flaminio Squazzoni; Bahar Mehmani; Francisco Grimaldo, 2022, ''Replication data for: Measuring the developmental function of peer review: A multi-dimensional, cross-disciplinary analysis of peer review reports from 740 academic journals'', https://doi.org/10.7910/DVN/D96G2I, Harvard Dataverse, V1.

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
