# Peer review of "Measuring the developmental function of peer review: a multi-dimensional, cross-disciplinary analysis of peer review reports from 740 academic journals"

_PeerJ, doi:10.7717/peerj.13539_

## Round 0.1 · original submission · Minor Revisions

Both reviewers found your paper to be interesting and valuable. Both also had a number of suggestions for improvements that should substantially enhance the paper. Please address all of the reviewers' comments.

·

Basic reporting

In general, it is a well-written paper that analyzes a large set of data. The research on the topic is growing and the authors should enhance the introduction including some other studies in order to present the existing literature in detail. The findings coming from studies on the attitudes of the reviewers, their focus concerning different parts of the manuscripts, the influence of anonymity/openness of the review on the review reports, the accelerated peer review during the pandemic and the influence on the reports, the external vs internal reviewers and their review reports, etc should be cited.

Experimental design

The study methods are well explained and sufficient details are provided.

Validity of the findings

I am not an expert in statistics. However, the conclusion is well stated deriving from the results.

Additional comments

Since the screening of the paper is the task of the editors/academic editors during the initial screen and before assigning reviewers, the major task of the Reviewers is to provide constructive feedback. Therefore the first sentence in the abstract is confusing.

A new term "developmental" in this area is introduced. 70-75 Authors mention that it refers to measuring the standards and practices of the review reports. It should be explicitly explained in the introduction what the term developmental contains.

Line 235-242
It seems like the impact factor of the journal does not affect the quality/development of the review reports. There are exceptions among SSE journals as well as journals in Physical and natural sciences. Not sure if this can support the conclusion that "the highest impact factor journals, in general, showed higher developmental standards of their review reports''. Table 11 for instance. Q3 and Q2 journals are positioned pretty high as well. For instance line 295-303 confirms that additional time does not affect the thoroughness of the reports.

243-246
Is the quality/constructiveness of the review reports the reason or the only reason for the delay in submitting a review report? Does it look like a fast and bold conclusion?? Many other factors usually affect the speed of the reviewers.

258-263
HMS and LS seem like a significant proportion of the journals that do not conform with the conclusion? Can this question the conclusion?

264-271
It refers to which parts of the paper reviewers value the most. It seems the results are in line with some recent studies focused only on this topic.

If funding is received a statement would be necessary.

Reviewer 2 ·

Basic reporting

1. There are several inconsistencies in language and writing.
2. The citations are not in the proper format.

Experimental design

1. More details are required on how information is extracted from peer reviews based on the 14 dimensions. What kind of techniques did you use to automatically extract the review statements that fall under those 14 developmental functions?
2. Need to define developmental functions elaborately.

Validity of the findings

1. Findings are interesting. It would be better that each finding is organized in sub-sections and then explained with data.

Additional comments

Nice large-scale study. Appreciate the efforts. The authors should consider making the data public for further research by the community. Also, the authors should cite some recent datasets on peer review analysis.

Reviewer 3 ·

Basic reporting

.

Experimental design

.

Validity of the findings

.

Additional comments

The paper is interesting, but lacks enough detail for the reader to really understand what is done. Therefore, several things need to be clarified.

General:
The claim in the beginning that peer review is the guarantee of trust in science seems overstated. Why would the opinion of two or three colleagues have such a weight. I guess this is just what many scientists believe, but I would guess that replication is the only guarantee. Of course the quality of peer review is important where peer review is decisive – as in acceptance of papers or selection of grants. However, at the same time, a lot of selection is not peer review at all. Grant selection panels are not peers, and in publication decisions, much is not peer review but desk decisions by editors or editorial staff.

Lacking details about the analysis:
• Line 94: how was this cleaning of the texts done?
• The test of the dimensions of Arcadia and the adaptation of the dimensions need more details of how it was done.
• Line 136-150: the description is very brief, and would not allow anybody to replicate the study. May be the details could be in an annex.
• Same holds for the description of the term selection (lines 151-157) and the dictionary building
• The resulting development score, is it based on the numbers of terms, or on the shares of terms?
• Section on ‘seniority’: how was the matching of the authors and scopus done on such a large scale.
• Also the disambiguation of reviewers <-> scopus. How was that done?
• Isn’t Cronbach alpha influenced by the number of items? If so, do the values of about 0.8 or higher say much, given the huge number of items?

Issues related to the findings
• Standards derived are derived from the data -> why would the empirical average be the correct one, and not e.g. the median?
• Figure 1 needs explanation on how to read it
• The tables with the statistical analysis (T10, T12, T13, T14) should give more details. For example, it is unclear what type of analyses are represented by the tables.

Issues related to the discussion
• Lines 291-294: what means ‘robust’ if it is not the same as ‘shared’? Could it also be shared and permissive?
• The discussion sections contains bivariate relations. Given the number of cases (reviews), a multivariate analysis would have been useful, and may show that some relations disappear after introducing relevant covariates.
• The discussion about the limitations is not very convincing, as many of the issues raised there seem not much related to the analysis in this paper. For example, why would one expect that a non-binary gender classification would influence the results at all? And, another example (line 368): “these editorial practices (of fast decisions) can influence - even if indirectly - the development of manuscripts in that drawing a straight line between quality screening and developmental function of peer review can be difficult Cowley (2015)”. One would like to read how this would influence the analysis, in order to assess the relevance of the analysis.

Some smaller issues
• Please put the Titles of the tables ABOVE the tables, and the titles of the figures BELOW the figures (this is the standard way of doing it)
• All areas of research: LS, PS, HMS, and SSE. -> Please add in the beginning of the paper what the abbreviations mean. This is now only done at page 8.
• Line 249: The reviewers OBTAINED a score?
• Line 264: Begin of the sentence is lacking
• Lines 287-291: sentence difficult to read due to the chaotic placement of the references.
• Line 337 and Line 346 both start with ‘Second’.

---

## Round 0.2 · accepted · Accept

Thank you for your careful attention to reviewers' comments.

·

Basic reporting

I agree with the current version.
However, in the discussion part is stated that women have shown more developmental tendencies in their reports BUT only in some areas. However, in the abstract is stated that this is "in general". The same for some other demographic factors.

Experimental design

Ok

Validity of the findings

ok

Additional comments

no